# Seven Fatty Acid Metabolism-Related Genes as Potential Biomarkers for Predicting the Prognosis and Immunotherapy Responses in Patients with Esophageal Cancer

**DOI:** 10.3390/vaccines10101721

**Published:** 2022-10-15

**Authors:** Ya Guo, Shupei Pan, Yue Ke, Jiyuan Pan, Yuxing Li, Hongbing Ma

**Affiliations:** Department of Radiation Oncology, The Second Affiliated Hospital, Xi’an Jiao Tong University, Xi’an 710004, China

**Keywords:** esophageal cancer (ESCA), The Cancer Genome Atlas (TCGA), fatty acid metabolism, prognosis, immunotherapy response

## Abstract

Background: Esophageal cancer (ESCA) is a major cause of cancer-related mortality worldwide. Altered fatty acid metabolism is a hallmark of cancer. However, studies on the roles of fatty acid metabolism-related genes (FRGs) in ESCA remain limited. Method: We identified differentially expressed FRGs (DE-FRGs). Then, the DE-FRGs prognostic model was constructed and validated using a comprehensive analysis. Moreover, the correlation between the risk model and clinical characteristics was investigated. A nomogram for predicting survival was established and evaluated. Subsequently, the difference in tumor microenvironment (TME) was compared between two risk groups. The sensitivity of key DE-FRGs to chemotherapeutic interventions and their correlation with immune cells were investigated. Finally, DEGs between two risk groups were measured and the prognostic value of key DE-FRGs in ESCA was confirmed in other databases. Results: A prognostic model was constructed based on seven selected DEG-FRGs. TNM staging and CD8+ T cells were significantly correlated with high-risk groups. Low-risk groups exhibited more infiltrated M0 macrophages, an activation of type II interferon (IFN-γ) responses, and were found to be more suitable for immunotherapy. Seven key DE-FRGs with prognostic value were found to be considerably influenced by different chemotherapy drugs. Conclusion: A prognostic model based on seven DE-FRGs may efficiently predict patient prognosis and immunotherapy response, helping to develop individualized treatment strategies in ESCA.

## 1. Introduction

ESCA is the eighth most common cancer worldwide causing the sixth highest cancer-related mortality rates [1]. Despite the range of available treatment options, the overall five-year survival rate remains less than 20%, with most deaths being associated with distant metastases and the emergence of resistance to chemoradiotherapy [2,3]. Therefore, exploration of novel therapeutic options and development of prognostic models for the management of ESCA are crucial.

The reprogramming of energy metabolism is a hallmark of cancer development, promoting cell growth and proliferation [4,5]. Accumulating evidence suggests that fatty acid metabolism plays a crucial role in metabolic reprogramming, affecting cell membrane formation, energy storage, and the production of signaling molecules [6,7]. Previous study has reported that fatty-acid-metabolism-related genes are associated with malignancy, prognosis, and immune phenotype in gliomas [8]. In cervical cancer patients, enhanced lipolysis and fatty acid synthesis promote lymphatic spread via the activation of nuclear factor kB (NF-kB) signaling [4,9]. Activated fatty acid oxidation improves the survival of acute myeloid leukemia cells [10]. In addition to influencing the effectiveness of chemo- and radiation therapy, altered fatty acid metabolism has also been suggested to effect responses to immunotherapy [11,12]. Previous study has indicated that fatty acids can affect the phenotype and functionality of infiltrating immune cells, potentially causing immunosuppression [13]. However, the prognostic relevance and potential therapeutic significance of genes involved in fatty acid metabolism, particularly in the context of immunotherapy, has never been explored among ESCA patients.

In this study, we identified differentially expressed FRGs (DE-FRGs). Then, the DE-FRGs prognostic model was constructed based on univariate Cox analysis, LASSO regression analysis, and multivariate Cox analysis. A receiver operating characteristic (ROC) curve was drawn to validate the accuracy of this model. Moreover, the correlation between the model and clinical characteristics was evaluated. A nomogram for predicting survival was established and evaluated. Subsequently, the difference in immune cell infiltration, immune function and immunotherapy response were compared between high- and low-risk groups. The sensitivity of key DE-FRGs to chemotherapeutic interventions and their correlation with immune cells were investigated. Finally, DEGs between two risk groups were identified and the prognostic value of key DE-FRGs in ESCA was confirmed in other databases.

## 2. Materials and Methods

### 2.1. Identification of Fatty-Acid-Metabolism-Related Genes

ESCA-related RNA sequencing data and corresponding clinical data, including information of age, gender, tumor stage, and survival information were downloaded from TCGA Database (https://portal.gdc.cancer.gov/ 15 March 2022). A total of 171 samples were obtained from TCGA database, including 160 tumor samples and 11 normal samples. A list of genes related to fatty acid metabolism (FRGs) was compiled using the Molecular Signature Database (MSigDB; https://www.gsea-msigdb.org/gsea/msigdb 14 March 2022), including the Kyoto Encyclopedia of Genes and Genomes (KEGG), the Reactome fatty acid metabolism, and the Hallmark fatty acid metabolism datasets, followed by the elimination of duplicate transcripts/genes [14]. The expression of fatty acid metabolism-related genes (FRGs) in ESCA were obtained from TCGA database using “Limma” R package. The differentially expressed fatty acid metabolism-related genes (DE-FRGs) in ESCA were identified using “Limma” and “pheatmap” R package. The criteria for identifying DE-FRGs were fold change (FC) >1.5 and false discovery rate (FDR) of <0.05 [15]. The details of clinical information can be found in Appendix A.

### 2.2. Construction and Evaluation of a Predictive Risk Score Model

Univariate Cox regression analysis was performed to identify DE-FRGs associated with prognosis, using the “survival” and “survminer” packages in R. Additional LASSO regression analysis was conducted to narrow further the field of key DE-FRGs. Subsequently, multivariate Cox regression analysis was performed to develop a prognostic risk score model for predicting overall survival (OS) of ESCA patients. ESCA patients were divided into high-risk (*n* = 80) and low-risk (*n* = 80) groups based on the median risk score. Kaplan–Meier (KM) survival curves were plotted to analyze the difference in overall survival (OS) and progression free survival (PFS) of patients assigned into the high-risk and low-risk groups. Finally, the receiver operating characteristic (ROC) curve was drawn and the area under the curve (AUC) for 1-, 3-, and 5-year OS was calculated through the “survival ROC” package in R to assess the predictive accuracy of the prognostic risk score model. A *p*-value < 0.05 was used as the filter condition.

### 2.3. The Association between Risk Score and Clinical Characteristics

The Limma R package was utilized to explore the association between risk score and clinical characteristics, including age, gender, and TNM pathology stage [14]. This was followed by univariate and multivariate Cox regression analyses to identify independent prognostic factors using the “survival” package (*p* < 0.05) [16].

### 2.4. Construction of a Nomogram for Patients with ESCA

A nomogram consisting of age, gender, pathologic staging, and prognostic risk score model was constructed using the “survival”, “regplot” and “rms” packages, to predict the likelihood of 1-, 3-, and 5-year survival of the studied ESCA patients. Then, independent prognostic analysis was performed to evaluate whether nomogram could independently predict patient prognosis. ROC curves and calibration curves were plotted to confirm the prediction accuracy of the nomogram [17].

### 2.5. Association between the Risk Model and Immune Parameters

The extent of immune cell infiltration was compared between the high- and low-risk groups using the CIBERSORT algorithms [18] and potential differences in immunologic functioning were evaluated using the “Limma”, “GSVA”, “GSEABase”, “ggpubr”, and “reshape2” R packages [19]. Finally, the tumor immune dysfunction and exclusion (TIDE) algorithm (http://tide.dfci.harvard.edu/login/ 18 March 2022) was applied to predict the effects of the observed changes on potential immunotherapy responses in the high- and low-risk groups [20].

### 2.6. GSCA Analysis 

Gene Set Cancer Analysis (GSCA, http://bioinfo.life.hust.edu.cn/GSCA/#/ 20 June 2022) is an online analysis tool for genomic, pharmacogenomic, and immunogenomic gene expression analyses in cancer [21]. We used GSCA to analyze the correlations between the expression of key biomarkers, immune cells, and drug sensitivity.

### 2.7. A Protein–Protein Interaction Network of DEGs in Groups of Different Risk Score Groups

DEGs in 160 ESCA patients between two different risk groups were analyzed using the Limma R package (|logFC| > 1, FDR < 0.05) to identify DEGs [4]. Gene Ontology (GO) enrichment analyses of the detected DEGs was then performed using the “clusterProfiler” R package [22]. A protein–protein interaction (PPI) network of DEGs was then constructed and visualized via the STRING database (STRING, https://string-db.org/ 17 March 2022) and the Cytoscape software (version: 3.7.2), identifying hub genes using cytoHubba [4]. According to the median expression value of the hub genes, all samples were divided into low- and high-expression groups. Kaplan–Meier analysis was carried out to compare the survival characteristics between the two groups [23]. Finally, the correlation between immune cell infiltration and prognostically relevant DE-FRGs was analyzed [24].

### 2.8. Validation of the Expression and Prognostic Value of Seven FRGs

UALCAN (http://ualcan.path.uab.edu/index.html 28 June 2022) is a comprehensive interactive web resource for analyzing cancer OMICS data (TCGA, MET500, CPTAC, and CBTTC), allowing users to identify biomarkers or to perform in silico validation of potential genes of interest. It uses graphical representations of gene expression profiles of protein-coding, miRNA-coding, and lincRNA-coding genes and combines these with survival information [25]. UALCAN database was used to confirm the expression difference of key DE-FRGs and investigate whether their expression was correlated with survival differences between the two groups.

## 3. Results

### 3.1. Identifying Fatty-Acid-Metabolism-Related DEGs in ESCA Samples

The database search-derived lists of fatty acid metabolism-related genes were intersected, and duplicates were eliminated. A total of 309 genes were identified as having a known or proposed role in fatty acid metabolism (Figure 1A). Of these, 108 were differentially expressed between ESCA samples and healthy samples in the TCGA database (Figure 1B,C; Table 1).

### 3.2. Establishing and Validating a Prognostic FRG Signature

Univariate Cox regression analysis indicated that 19 differentially expressed DE-FRGs (DE-FRGs) were associated with different clinical outcomes (*p* < 0.05) (Figure 2A). LASSO analyses were used to further analyze 19 DE-FRGs to preventing overfitting. This additional analysis identified a signature, consisting of 11 genes, that showed altered expression levels depending on clinical prognosis (Figure 2B,C). Subsequently, seven DE-FRGs were identified as potential prognostic-related biomarkers of ESCA patients using multivariate Cox regression analysis (Figure 2D). Kaplan–Meier analysis indicated that compared with the low-risk group, the high-risk group was significantly associated with poor OS and PFS (*p* < 0.05) (Figure 2E,F). Furthermore, time-dependent ROC curves were constructed and the corresponding area under the curve (AUCs) figures were calculated to assess the predictive power of the prognostic risk model. The AUC values for 1-, 3-, and 5-year OS were 0.787, 0.829, and 0.937, respectively (Figure 2G).

### 3.3. Association between Risk Score and Clinical Characteristics

We analyzed the association between patient clinical characteristics and the gene expression-based risk score. This analysis showed no association between the age or gender of the patients and their corresponding risk scores (Figure 3A,B). However, a higher risk score correlated with significantly more advanced pathologic stage and T stage of the tumor (*p* = 0.0017, Figure 3C, Appendix A). Univariate and multivariate Cox regression analysis revealed that our risk score and the clinical T stage could serve as independent prognostic factors (*p* < 0.001) (Figure 3D,E). ROC curve analysis further supported the high sensitivity and specificity of the predictive score (Figure 3F,G).

### 3.4. Establishing and Evaluating a Nomogram for Predicting Survival

A nomogram, integrating age, gender, TNM stage of the tumor, and the risk score model was established to predict OS in ESCA patients (Figure 4A). The calibration curves at 1 year, 3 years, and 5 years indicated that the nomogram could accurately predict the OS of ESCA patients (Figure 4B). Based on univariate Cox regression analysis the nomogram model and the TNM tumor stage could predict the prognosis of the patients independently of each other or other clinical parameters (Figure 4C). Multivariate Cox regression analysis showed that the nomogram model was an independent prognostic factor with an HR of 1.193 (95% CI = 1.060–1.343) (Figure 4D). ROC analysis also revealed that the nomogram could predict the OS for ESCA patients with remarkable accuracy (AUC: 1 year = 0.736, 3 years = 0.849; Figure 4E,F).

### 3.5. Immunological Features of the Tumor and GSCA Analysis

The expression of seven key FRGs showed strong correlation with the infiltration of the tumors with immune-activating cells (Figure 5A). The ratios of 22 distinct immune cell types in the tumors of high- and low-risk ESCA patients is shown in Figure 5B. The number of infiltrating CD8+ T cells was higher in the high-risk group, while M0 macrophages were more predominant in the low-risk patients (Figure 5B). In terms of immune function, our results also showed that type II IFN production was significantly higher in the low-risk group (Figure 5C). Furthermore, significantly higher TIDE scores were seen in the low-risk than group, indicating that potentially the high-risk group could benefit more from receiving immunotherapy (Figure 5D). We also explored the correlation between the expression of the seven key FRGs and drug sensitivity, defined as IC50 values, based on Spearman’s correlation analysis. Our results revealed that most drugs effected the seven key FRGs, suggesting that these molecules could be exploited as potential therapeutic drug targets in the management of ESCA (Figure 5E,F).

### 3.6. DEGs in the Low- and High-Risk Score Groups

Using the “Limma” package, we identified 48 genes that were differentially expressed between the high- and low-risk groups (Table 2). GO analyses indicated that these DEGs mainly belonged to the response to glucocorticoid, response to corticosteroid, intermediate filament cytoskeleton organization, intermediate filament-based process, skin development, negative regulation of peptidase activity, unsaturated fatty acid biosynthetic process, and regulation of peptidase activity KEGG pathways (Figure 6A). A PPI network of 48 DEGs was constructed using the STRING database (Figure 6B). The Top 10 genes of the network, including PPL, MMP9, TGM1, ALOX12, ANXA1, VIL1, IL1RN, GPA33, MLXIPL, and PCK1 of the network were selected using the cytoHubba plugin in Cytoscape (Figure 6C). Analyzing these against the survival parameters showed that high expression of PPL was significantly associated with favorable OS of ESCA patients (Figure 6D). The proportions of infiltrating immune cells in samples expressing PPL at high or low levels was analyzed using the “Limma” package. Our result showed that the presence of M0 macrophages was significantly more pronounced in samples with a high PPL high expression. In addition, the proportion of T cell CD4+ resting memory T cells was lower in the immune infiltrates in the high-PPL-expression group (Figure 6E).

### 3.7. Validating the Expression of the Seven FRGs and Their Prognostic Value

The UALCAN online database was used to analyze how the expression of key FRGs affected survival times. The results indicated that FABP2, HSPH1, and IDH3G were more abundantly expressed in tumors (Figure 7A–C). Furthermore, a higher abundance of PDHA1 correlated with poor prognosis in ESCA patients (Figure 7D). Similarly, the increased expression of HSPH1, IDH3G, NUDT7 and PDHA1 was associated with shorter OS in the subgroup of patients suffering from esophageal adenocarcinoma (EAD) (Figure 7E–H).

## 4. Discussion

There is increasing evidence that metabolic dysregulation plays a critical role in cancer cell growth, proliferation, angiogenesis, and invasiveness [14,26]. Previous observations suggest that abnormal glycolytic metabolism is associated with the physiological behavior of human malignant tumors [27]. In colorectal cancer, abnormal anaerobic metabolic pathways play an important role in the formation of cancer stem-like cells (CSCs), promoting the rapid formation, development, and therapy resistance of these tumors [28,29]. Fatty acid metabolism is involved in cellular energy production, membrane synthesis, and signal transduction pathways relevant to tumorigenesis and development [14,30]. Deregulated anabolism/catabolism of fatty acids may support cancer cell growth [6]. A previous study reported that fatty acid synthesis may promotes esophageal adenocarcinoma [31]. A recent study has shown that loss of FBP1 promotes migration, proliferation and invasion through regulating fatty acid metabolism in ESCA [32].Although several studies focused on the role of fatty acid metabolism in a variety of tumors, this topic remains unclear in ESCA. The identification of key molecular markers related to fatty acid metabolism, together with the exploration of their role in the development of ESCA, could provide novel insights into the biological behavior of these tumors, potentially highlighting new, more effective, therapeutic strategies. 

In the present study, we explore the role of DE-FRGs in ESCA. Interestingly, 19 FRGs that were differentially expressed in patients with ESCA showed a correlation with the clinical/biological behavior of the tumors. Via LASSO and multivariate Cox regression analysis, we were able to identify seven key DE-FRGs, including PDHA1, CD36, IDH3G, HSPH1, FABP2, NUDT7 and SERINC1 that may become useful prognostic biomarkers in the clinical management of ESCA patients. Subsequently, a prognostic risk score model was established that was able to divide patients into distinct high- and low-risk groups showing significant differences in OS and DFS. This prognostic risk score was an independent prognostic factor according to univariate and multivariate Cox regression analyses. Furthermore, the predictive potential of this model was confirmed by combination with clinical characteristics (age, gender, and TNM stage of the tumor) in a risk-assessment nomogram. The risk model presented here might help to identify ESCA patients with a poor prognoses, and could be utilized in the management of the disease. 

PDHA1(Pyruvate Dehydrogenase E1 Subunit Alpha 1) is a protein-coding gene involved in the pyruvate and thiamine metabolism pathways. In HNSCC cells, LDHA/PDHA1 changes may associated with a broad metabolic reprogramming while intracellular molecules including polyunsaturated fatty acids and nitrogen-metabolism-related metabolites underlie the malignant changes [33]. Previous study has shown that promotion of the tricarboxylic acid cycle (upregulation of PDHA1, PDHB, ACO2, and DLST expression) and inhibition of ME1 expression may inhibit fatty acid synthesis [34]. The inhibition of PDHA1 expression in the LnCap human prostate cancer cells led to the “Warburg effect”, resistance to chemotherapy, improved migration, and increased expression of stem cell markers [35]. In ESCC, the low expression of PDHA1 correlates with poor clinical prognosis and can result in metabolic reprogramming, again leading to the Warburg effect increasing malignant potential [36,37]. Decreased PDHA1 protein expression was also found to predict poor prognosis in gastric cancer [38]. HSPH1 encodes a member of the heat shock protein 70 family. Previously, elevated levels of HSPH1 expression were reported in tissues of HNSC patients. Moreover, the overexpression of HSPH1 was associated with poor overall survival (OS) [39]. HSPH1 is one of the most prominently upregulated proteins in several malignancies, with a well-documented involvement in Wnt- and chronic nuclear factor-kappa B signaling [40,41,42]. It has been suggested that the analysis of HSPH1 expression levels may help in predicting the effectiveness of chemotherapeutic approaches acting on the EGFR-TKI pathway in advanced lung adenocarcinoma [40]. The involvement of HSPH1 in anticancer immunity has also been described [41]. NUDT7, Nudix Hydrolase 7, is an enzyme involved in peroxisomal lipid metabolism. In the liver, upregulated expression of NUDT7 can inhibit peroxisomal fatty acid oxidation [43]. In clear cell renal cell carcinomas the alternative splicing of NUDT7 was found to be correlated with overall survival time [44]. Fatty acid binding proteins (FABPs) are key proteins in lipid transport, which can maintain a steady pool of fatty acids in the epithelium by traffic lipids from the intestinal lumen to enterocytes and bind superfluous fatty acids. As a lipid chaperone, FABP2 can also carry lipophilic drugs to improve targeted transport [45]. During the work presented here we identified and validated the prognostic value of seven key DE-FRGs. Of these FABP2, HSPH1, and IDH3G were upregulated and the high expression of four FRGs, including PDHA1, HSPH1, IDH3G, and NUDT7 was associated with poorer OS. These findings suggest that these key DE-FRGs might have a prognostic value in the assessment of ESCA and could represent potential therapeutic targets. 

Previous studies have suggested that prominent CD8+ T cell infiltrates were associated with clinical prognosis and immune responses in ESCA [46,47]. In renal cell cancer an increased CD8+ T-cells to Treg ratio is associated with poor prognosis [48]. We observed an increase in the number of infiltrating CD8+ T cells in the samples of high-risk group ESCA patients, indicating an unfavorable prognosis. In contrast, patients with a low-risk score showed an activated type II IFN response. Based on the TIDE algorithm this may indicate suitability for the immunotherapy. Previous studies demonstrated that type II IFN (IFN-γ) could indirectly regulate PD-L1 levels in small cell lung cancer [49]. It was also suggested that IFN-γ production induced elevated PD-L1-mRNA expression resulting in favorable OS [50]. Indeed IFN-γ production is a key driver of PD-L1 expression in both cancer and host cells, and may improve the likelihood of anti–PD-1 therapies being effective [51]. These previous findings support the notion that the favorable prognosis of patients in the low-risk group may be due to more effective immune responses.

It was reported that the expression of PPL was significantly decreased in esophageal cancer tissues and that the proteins were barely detectable in advanced cancer samples [52]. The experimental knockdown of PPL decreased cellular motility, reduced attachment, and generally inhibited malignant progression [53]. However, this finding contrasts with observations that the high expression of PPL is associated with favorable survival in patients with adenoid cystic carcinomas and sarcomas [54]. Our results showed that the expression of PPL gene was upregulated in low-risk group, and high expression of this gene was associated with better prognoses. This result may further support the hypothesis that the low-risk group was significantly associated with a favorable prognosis.

## 5. Conclusions

In summary, this first investigation of the role of fatty acid metabolism-related genes in ESCA identified seven genes showing strong association with the prognosis and therapeutic responses to chemotherapy and immunotherapy in this disease. This allowed us to develop a risk score model that could effectively divide patients into high- and low-risk groups. Using this score clinically might aid the development of individualized treatment strategies in the future. However, the prognosis value and immunotherapy effect of seven fatty-acid-metabolism-related genes in patients with esophageal cancer need to be further confirmed in larger clinical studies.

## Figures and Tables

**Figure 1 vaccines-10-01721-f001:**
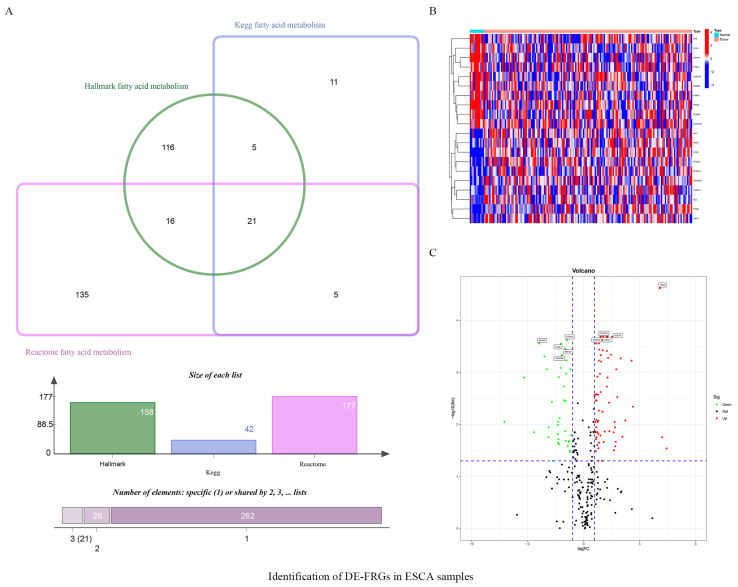
Identification of DE-FRGs in ESCA samples. (**A**) 309 fatty acid metabolism-related genes from Kegg, Hallmark and Reactome. (**B**) Heatmap display the upregulated and downregulated fatty acid metabolism−related DEGs (top 10, respectively, *p* < 0.05). (**C**) Volcano plot of fatty acid metabolism-related DEGs (only displaying 10 DE-FRGs).

**Figure 2 vaccines-10-01721-f002:**
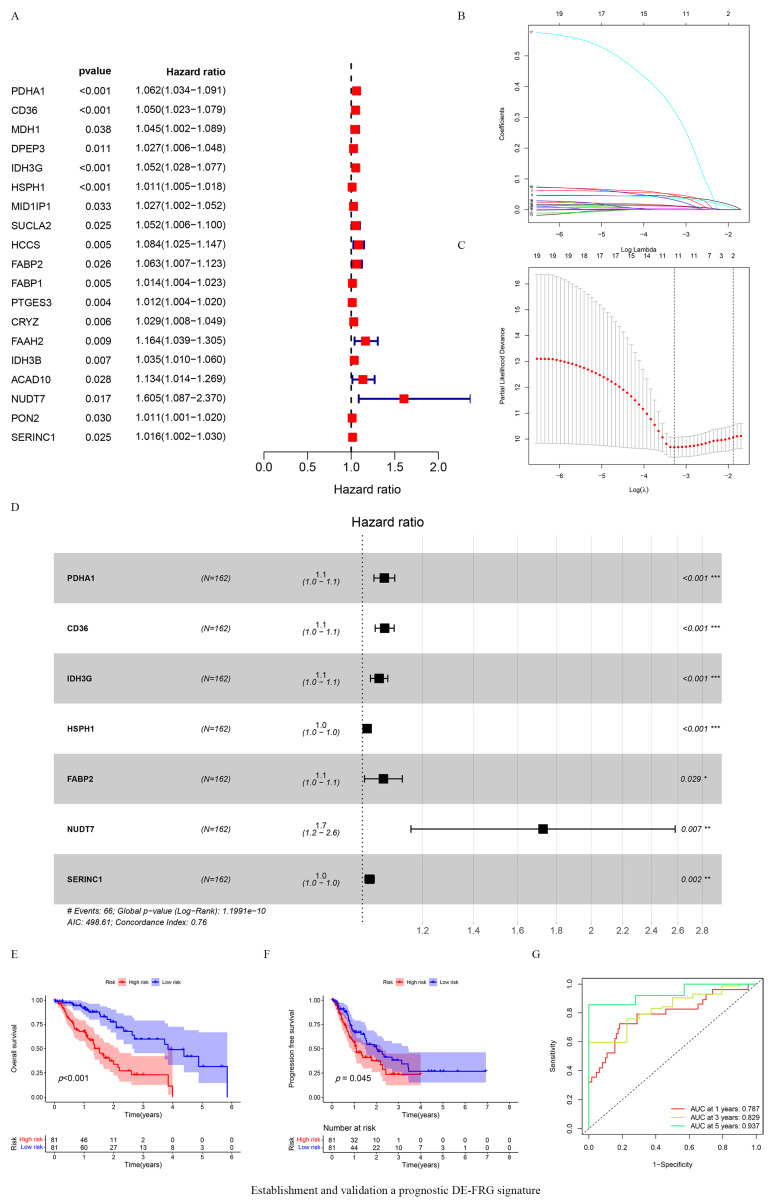
Establishment and validation a prognostic DE-FRG signature. (**A**) 19 genes were associated with the clinical outcomes of ESCA patients based on univariate Cox regression analysis; Red squares represnt Hazard Ratio (HR) value. (**B**) LASSO coefficient profiles of the 19 selected fatty acid metabolic genes; (**C**) The best parameter (lambda) in the LASSO-Cox model; the red dots indicates the partial probability of deviation values, the gray lines indicates standard error (SE). (**D**) 7 genes were eventually identified as prognosis related biomarkers based on multivariate Cox regression analysis; ESCA patients were divided into high-risk *(n* = 80) and low-risk (*n* = 80) groups based on the median risk score. * *p* < 0.05, ** *p* < 0.01, *** *p* < 0.001. Black squares represent Hazard Ratio (HR) value. (**E**,**F**) Kaplan–Meier survival analysis was performed to assess the difference in OS and PFS between the high-risk and low-risk group; (**G**) ROC curves of risk model for predicting overall survival at 1, 3, and 5 years.

**Figure 3 vaccines-10-01721-f003:**
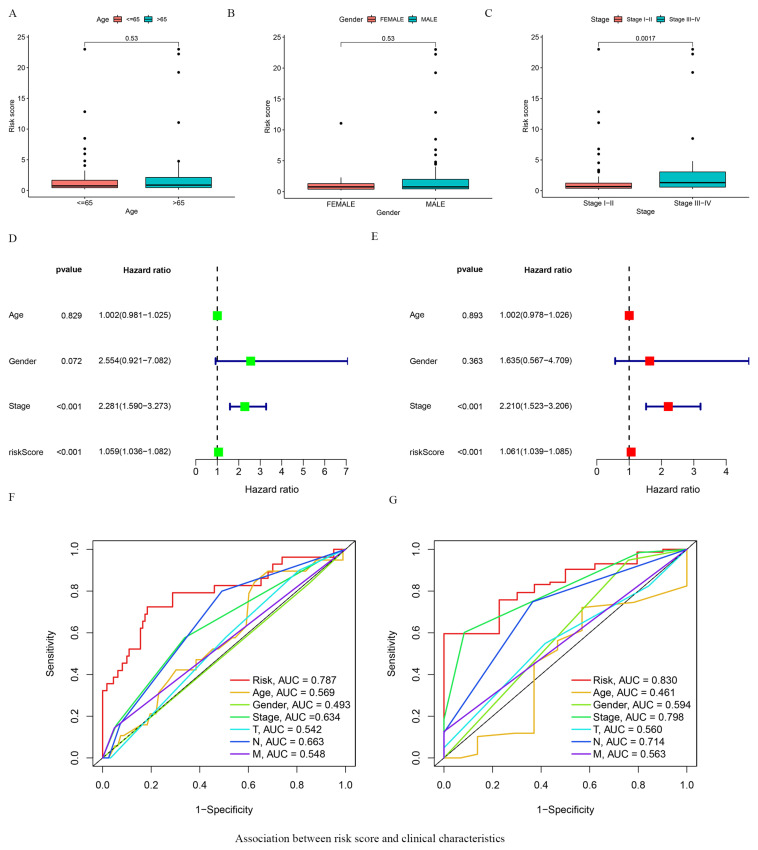
Association between risk score and clinical characteristics. (**A**–**C**) The association of risk score and clinicopathological features, including age (**A**), gender (**B**), and tumor stage (**C**); The circle represent each individual. (**D**) The univariate Cox regression analysis of clinical parameters in patients with ESCA; (**E**) The multivariate Cox regression analysis of clinical parameters in patients with ESCA; (**F**) ROC curves show 1-year survival in ESCA patients based on risk score and multiple clinical features; (**G**) ROC curves display 3-year survival in ESCA patients based on risk score and multiple clinical features. We excluded patients with deficient clinical information and 124 patients were retained for analysis.Red and green square represent HR value.

**Figure 4 vaccines-10-01721-f004:**
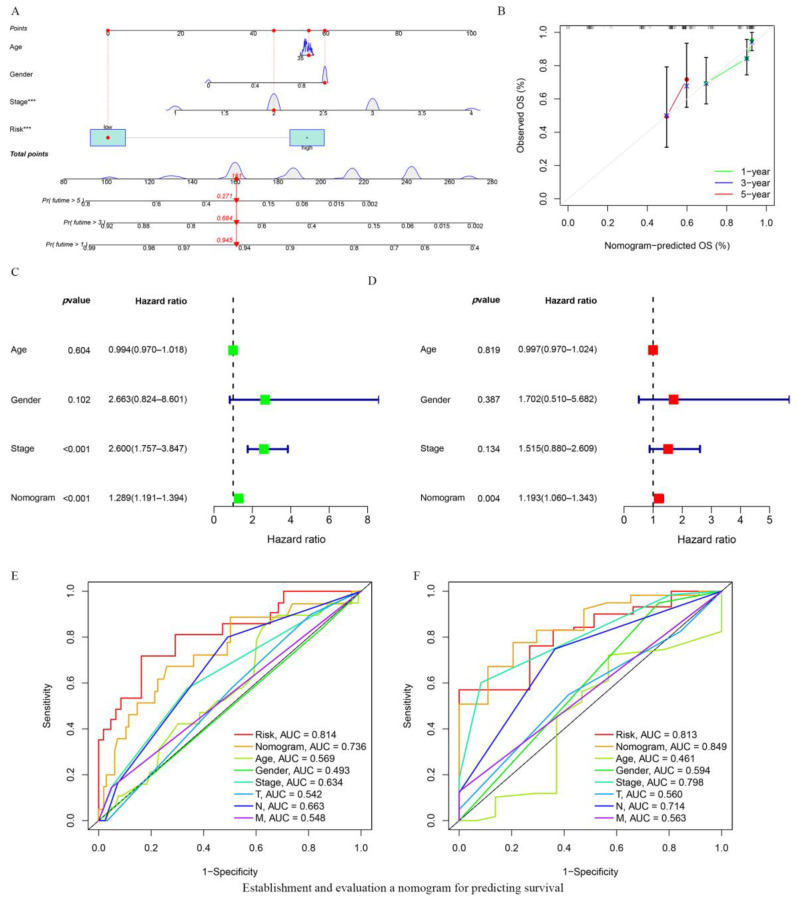
Establishment and evaluation a nomogram for predicting survival. (**A**) Nomogram predicting OS of ESCA patients. (**B**) The calibration plots of the nomogram. The x axis is nomogram-predicted survival, and the y axis is actual survival. (**C**) Univariate Cox regression analysis of the nomogram. (**D**) Multivariate Cox regression analysis of the nomogram. Red and green square represent HR value. (**E**,**F**) ROC curves at 1 and 3 years showing the predictive ability of the nomogram.

**Figure 5 vaccines-10-01721-f005:**
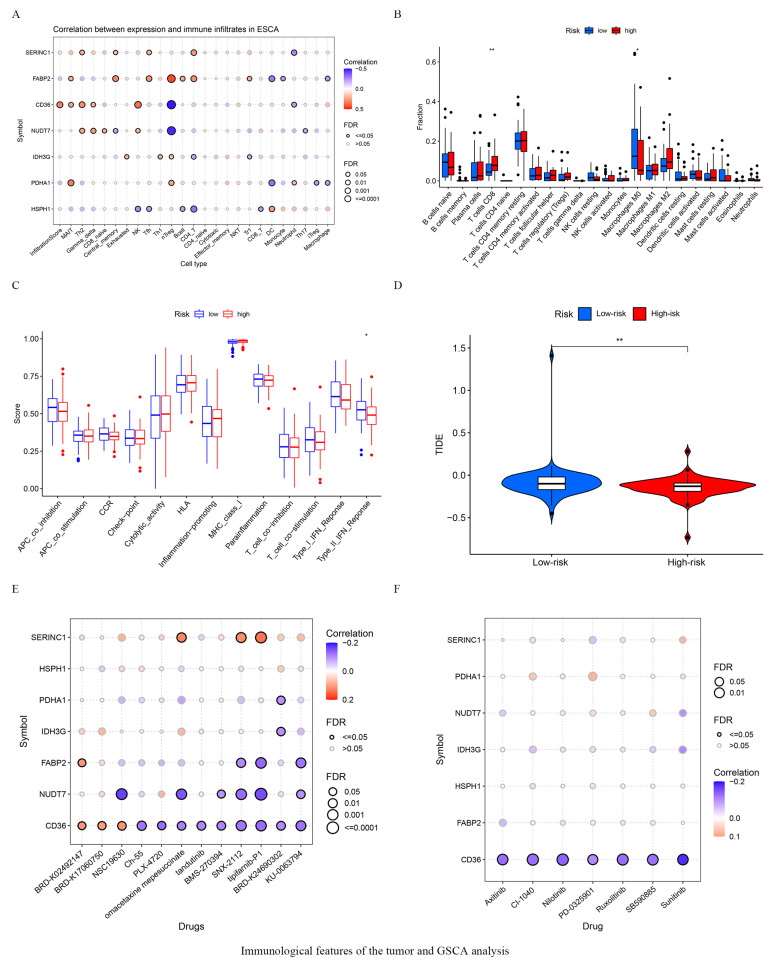
Immunological features of the tumor and GSCA analysis. (**A**) The correlation between 7 key genes and infiltrated immune cells. Different color circle denote different *p* value, the smaller the *p* value, the larger the circle. (**B**) The ratios of 22 immune cell types in high- and low-risk patients. The black circle represents a single individual. (**C**) Immune function difference in high- and low-risk group in ESCA patients. (**D**) Comparison of Immunotherpy response between high- and low-risk patients. (**E**) Correlation between CTRP drug sensitivity and 7 key gene expressions. (**F**) Correlation between GDSC drug sensitivity and 7 key gene expressions. * *p* < 0.05, ** *p* < 0.01.

**Figure 6 vaccines-10-01721-f006:**
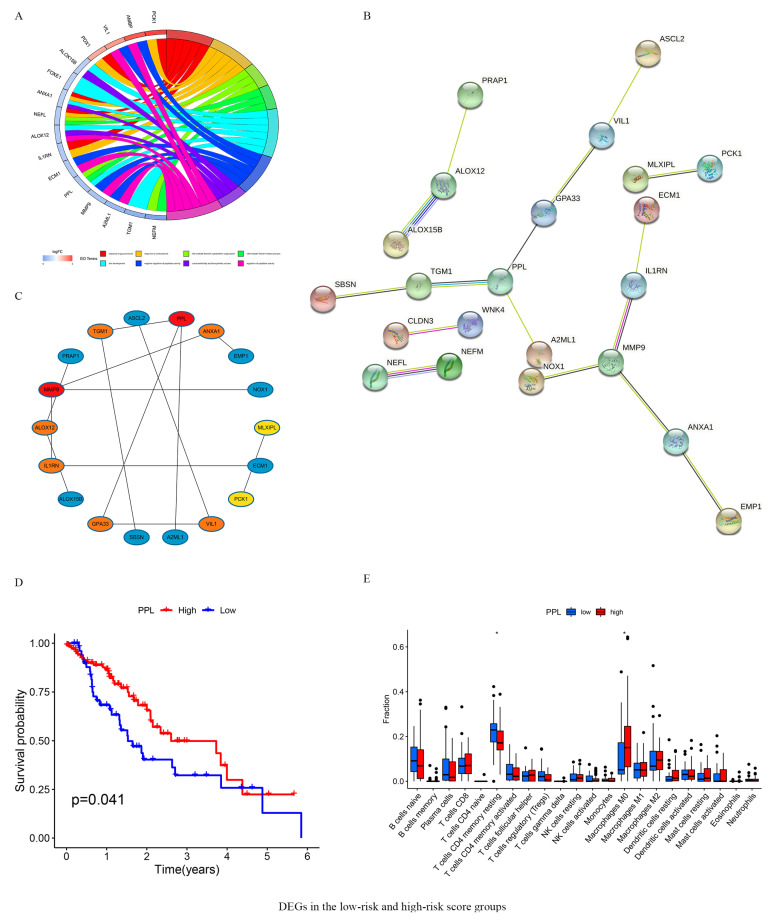
DEGs in the low- and high- risk score groups. (**A**) The results of GO enrichment analysis on DEGs. (**B**) PPI network processed by STRING. (**C**) Identification of 10 hub DEGs using cytoHubba software. (**D**) Survival analysis for subgroup patients stratified by PPL mRNA expression. (**E**) The abundance of each infiltrated cell in patients with high- and low PPL mRNA expression. DEGs in 160 ESCA patients between 2 different risk groups were analyzed using the Limma R package (|logFC| > 1, FDR < 0.05). Black circle represent individual values. * *p* < 0.05.

**Figure 7 vaccines-10-01721-f007:**
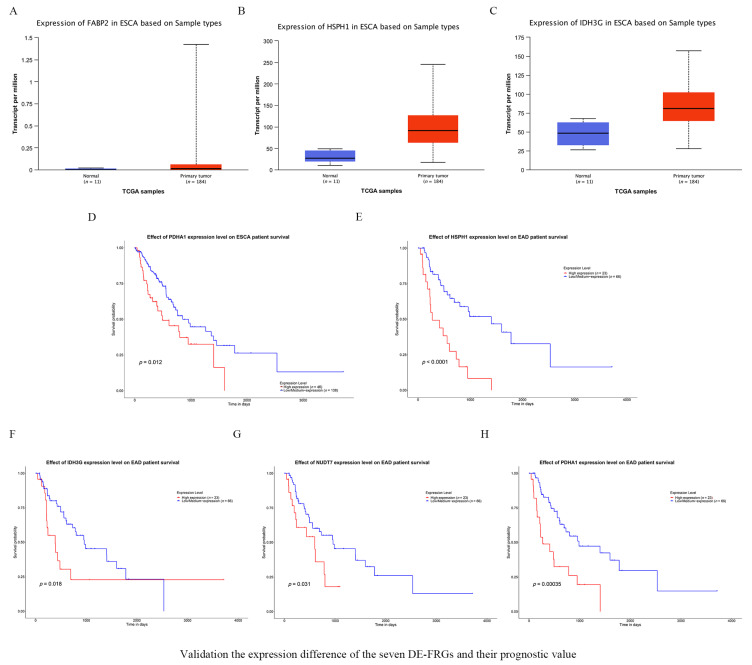
Validation the expression difference of the seven DE-FRGs and their prognostic value. (**A**–**C**) Confirmation of the expression of FRGs in ESCA. (**A**) FABP2, (**B**) HSPH1 and (**C**) IDH3G. (**D**) High PDHA1expression was correlated with poor OS in ESCA. (E–H) Survival rates were calculated between high and low gene expression groups in EAD patients. (**E**) HSPH1, (**F**) IDH3G, (**G**) NUDT7 and (**H**) PDHA1. EAD = Esophageal adenocarcinoma.

**Table 1 vaccines-10-01721-t001:** Identification of differentially expressed fatty acid metabolism-related genes.

Gene	logFC	*p*-Value	FDR
FADS2	1.6647	0.0022	0.0090
HACD1	−0.8386	0.0109	0.0258
PTGES2	0.8793	0.0001	0.0011
HPGD	−1.9718	0.0001	0.0009
PTGES	2.5728	0.0001	0.0006
MAOA	−0.9333	0.0001	0.0011
CD36	−0.6685	0.0053	0.0161
ELOVL2	1.7260	0.0084	0.0217
GPX1	0.6824	0.0031	0.0114
ENO3	0.7959	0.0125	0.0288
CEL	4.4503	0.0128	0.0291
ACBD6	0.8421	0.0000	0.0004
MORC2	0.9267	0.0000	0.0002
UBE2L6	1.4763	0.0000	0.0005
CYP2C9	−1.3856	0.0050	0.0153
SLC25A1	0.6648	0.0006	0.0034
ALDH1A1	−1.6541	0.0256	0.0499
PRKAA2	−1.3722	0.0091	0.0227
ACOT7	1.5324	0.0000	0.0002
ABCD1	1.0133	0.0012	0.0057
GLUL	−1.6938	0.0007	0.0037
ACOT9	0.6684	0.0071	0.0193
NCAPH2	0.8264	0.0000	0.0003
HACD2	0.7813	0.0004	0.0027
OSTC	0.5871	0.000	0.0028
FADS1	1.6133	0.0046	0.0142
HMGCS2	−0.9822	0.0038	0.0136
ALOX15	4.2060	0.0063	0.0177
CPOX	0.9419	0.0001	0.0006
ACSM6	−2.6558	0.0043	0.0142
NTHL1	0.9232	0.0000	0.0005
MIX23	1.0243	0.0000	0.0004
HSPH1	1.3146	0.0000	0.0005
PTGDS	−0.7081	0.0160	0.0339
CYP2C8	−4.2436	0.0021	0.0089
PTGES3	1.0439	0.0000	0.0002
CPT1B	0.9034	0.0055	0.0163
ADH1C	−1.3389	0.0071	0.0193
BLVRA	0.6532	0.0152	0.0327
SUCLG2	−1.3197	0.0002	0.0012
THRSP	−1.3148	0.0099	0.0242
ACOX3	−0.5875	0.0115	0.0266
ELOVL3	2.6467	0.0009	0.0046
TDO2	4.0796	0.0000	0.0000
ACADS	−0.9901	0.0007	0.0035
YWHAH	0.7439	0.0002	0.0012
AMACR	−0.7589	0.0080	0.0209
PCCA	−0.9226	0.0001	0.0006
ODC1	1.8059	0.0005	0.0030
ALAD	−1.2068	0.0000	0.0003
CA2	−1.8684	0.0031	0.0114
RDH11	0.6085	0.0004	0.0027
LDHA	1.2471	0.0003	0.0019
ACADSB	−1.1380	0.0000	0.0005
ACAT1	−1.2122	0.0001	0.0008
GABARAPL1	−0.9593	0.0080	0.0209
ALOX5AP	1.0724	0.0086	0.0219
NSDHL	0.7857	0.0020	0.0085
FMO1	2.2368	0.0019	0.0084
ACAT2	0.7943	0.0049	0.0152
GPX4	0.6893	0.0018	0.0082
ACSS1	−0.7269	0.0150	0.0324
ETFDH	−0.8984	0.0000	0.0002
ACSM3	−1.3758	0.0026	0.0102
ACBD7	1.6988	0.0113	0.0266
AUH	−0.8033	0.0083	0.0215
H2AZ1	1.2866	0.0000	0.0002
SMS	0.9141	0.0004	0.0024
ELOVL5	1.3837	0.0037	0.0135
ALDOA	0.7531	0.0023	0.0090
CYP4B1	−1.2830	0.0017	0.0076
ACO2	−0.7125	0.0001	0.0009
MDH2	0.6924	0.0004	0.0026
PPT1	1.4317	0.0000	0.0002
PON2	1.7812	0.0000	0.0002
PSME1	0.9799	0.0000	0.0002
PTGS2	1.6912	0.0031	0.0114
HSD17B10	0.6922	0.0007	0.0035
ACLY	1.2460	0.0000	0.0004
HMGCS1	0.8293	0.0089	0.0225
ECI2	−1.3581	0.0021	0.0089
METAP1	0.6304	0.0040	0.0138
APEX1	0.8167	0.0001	0.0006
MIF	1.2857	0.0002	0.0013
ADSL	0.9045	0.0000	0.0002
SCD	1.3604	0.0007	0.0038
RDH16	1.6334	0.0139	0.0315
PRXL2B	0.7951	0.0256	0.0499
IL4I1	1.9874	0.0000	0.0005
ACADL	−3.1819	0.0002	0.0013
PTGIS	−1.8784	0.0062	0.0177
ACAD11	0.7438	0.0103	0.0249
ACACB	−2.0952	0.0000	0.0005
ADH1B	−2.3779	0.0000	0.0003
ENO2	1.1322	0.0040	0.0138
NBN	0.6371	0.0022	0.0090
LGALS1	1.7481	0.0001	0.0009
GAPDHS	2.0802	0.0059	0.0173
SLC27A2	0.9832	0.0256	0.0499
HSP90AA1	1.2248	0.0000	0.0002
MLYCD	−0.9958	0.0000	0.0003
SLC25A17	0.6682	0.0000	0.0003
BCKDHB	−1.0404	0.0006	0.0034
DBI	0.6881	0.0062	0.0177
GPD2	0.6431	0.0045	0.0142
S100A10	0.7390	0.0046	0.0142
HSD17B7	0.8224	0.0011	0.0054
CYP4A11	−1.0684	0.0003	0.0019

**Table 2 vaccines-10-01721-t002:** Differentially expressed genes between high and low-risk groups.

Gene	logFC	*p*-Value	FDR
KRT16P2	−1.1383	0.0004	0.0175
RNF225	−1.5708	0.0038	0.0488
FOXE1	−1.1111	0.0035	0.0477
USH1G	−1.1662	0.0024	0.0386
BCAT1	−1.1360	0.0015	0.0301
MIEN1	1.1579	0.0012	0.0275
MLXIPL	1.2968	0.0012	0.0270
PPL	−1.3393	0.0006	0.0204
TMEM74B	1.0136	0.0003	0.0142
NEFM	−2.1322	0.0017	0.0325
GBP6	−1.4412	0.0018	0.0334
KRT16P6	−1.1958	0.0025	0.0398
ALOX15B	−1.1045	0.0008	0.0223
AHNAK2	−1.0332	0.0002	0.0111
AMBP	2.2574	0.0031	0.0450
MIR559	1.1306	0.0013	0.0275
ALOX12	−1.2123	0.0007	0.0212
PCK1	2.4876	0.0000	0.0039
PDX1	1.2546	0.0005	0.0187
PINLYP	−1.1081	0.0003	0.0151
FAM83C	−1.0035	0.0034	0.0471
ANXA1	−1.1213	0.0006	0.0199
YBX2	1.1975	0.0000	0.0047
MIR3189	1.1694	0.0028	0.0425
NOX1	1.1536	0.0002	0.0125
MMP9	−1.3834	0.0012	0.0272
EMP1	−1.1129	0.0009	0.0241
SBSN	−1.0531	0.0030	0.0439
CCL15	1.6283	0.0036	0.0480
ACY3	1.2213	0.0003	0.0153
ASCL2	1.4926	0.0034	0.0467
IL1RN	−1.2627	0.0002	0.0138
A2ML1	−1.6120	0.0008	0.0233
TGM1	−1.8265	0.0014	0.0295
NEFL	−1.1713	0.0006	0.0200
CLRN3	1.2620	0.0028	0.0427
QPRT	1.2541	0.0013	0.0275
VIL1	1.4509	0.0032	0.0454
WNK4	1.2133	0.0034	0.0469
GOLT1A	1.2505	0.0006	0.0206
ANKRD33B	−1.0516	0.0016	0.0314
CLDN3	1.4046	0.0004	0.0170
PRAP1	1.5993	0.0020	0.0354
ZBED2	−1.3404	0.0001	0.0100
ECM1	−1.2862	0.0022	0.0368
GPA33	1.5683	0.0023	0.0380
BCAN-AS1	1.1560	0.0004	0.0175
RNF157	1.2480	0.0011	0.0262

## Data Availability

The raw data of this study were obtained from the TCGA database (https://portal.gdc.cancer.gov/), which is a publicly available database. FRGs are derived from the Molecular Signature Database (MSigDB; https://www.gsea-msigdb.org/gsea/msigdb).

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
