# Peer review of "Seven Fatty Acid Metabolism-Related Genes as Potential Biomarkers for Predicting the Prognosis and Immunotherapy Responses in Patients with Esophageal Cancer"

_vaccines, 2022, doi:10.3390/vaccines10101721_

Round 1

Reviewer 1 Report

This paper investigates the role of fatty acid metabolism-related genes in esophageal cancer with the construction of prognostic models. On my behalf the paper is interesting to the biomed community and is worth publishing for two reasons: 1) showing the effectiveness of the new prognostic model proposed in obtaining valuable information on the patient prognosis and in therapy responses. 2) adding data (prognostic-biomarkers) on the prognosis of esophageal cancer. 

Despite the originality of the work, I would like the authors to expand on the following points:

-I have difficulties in the interpretation of some figures: please put the captions under the figures; the resolution of some figures must be improved (see fig. 1); include the name of the best DEGs (based on FC and p.val) in the volcano plot, please use software that allow a better representation  (eg. MetaboAnalyst or GraphPad); improve the quality of hierarchical clustering analysis (heat map) or use another test, this is not the best representation, I suggest to show the top DEGs and not all the significant DEGs; included a combined multiROC curve with all the risk factors, taking into account the all the possible variables. 

-please cite all the figures in the text, not just in the captions. 

-please, include a table with the clinical features of the patients involved in the study, healthy patients seem to be few. It is ok to include the table as supplementary materials). 

-implement also the discussion of results obtained with the identification of  fatty acid metabolism-related DEGs. Fatty acid metabolism is always cited but never discussed. Please Include a list of the main fatty acids related with the differentially expressed genes and comment on it. 

Author Response

This paper investigates the role of fatty acid metabolism-related genes in esophageal cancer with the construction of prognostic models. On my behalf the paper is interesting to the biomed community and is worth publishing for two reasons: 1) showing the effectiveness of the new prognostic model proposed in obtaining valuable information on the patient prognosis and in therapy responses. 2) adding data (prognostic-biomarkers) on the prognosis of esophageal cancer. 

Replay: Thank you very much for your professional evaluation and comments.

Despite the originality of the work, I would like the authors to expand on the following points:

-I have difficulties in the interpretation of some figures: please put the captions under the figures;

Replay: Thank you very much for your suggestion. We have added the captions under the figures according to the your suggestion.

The resolution of some figures must be improved (see fig. 1); include the name of the best DEGs (based on FC and p.val) in the volcano plot, please use software that allow a better representation  (eg. MetaboAnalyst or GraphPad);

Replay: Thank you for pointing this out. We have added the best DEGs (based on FC and p.val) in the volcano plot according to the your suggestion.

Improve the quality of hierarchical clustering analysis (heat map) or use another test, this is not the best representation, I suggest to show the top DEGs and not all the significant DEGs;

Replay: Thank your for your professional comment. We displayed the top 10 up-regulated and down-regulated expressed genes respectively in Figure 1B according to the your suggestion.

included a combined multiROC curve with all the risk factors, taking into account the all the possible variables. 

Replay: You have raised an important question. We gratefully appreciate for your valuable comment. We have added more clinical feature (such as T, N, M stage) to draw multiROC curve (Figure 3F-G, Figure 4E-F).

-please cite all the figures in the text, not just in the captions. 

Replay: Thank you for reviewing our manuscript carefully. We have added related content in the result section, figure legend and supplement material (Figure S1) according to the your suggestion.

-please, include a table with the clinical features of the patients involved in the study, healthy patients seem to be few. It is ok to include the table as supplementary materials). 

Replay: You have raised an important question. We are very grateful for your valuable comments. We have added the clinical features of the patients in the Materials and Methods section (line 80-81) and supplementary materials (Supplementary Table 1 and Table 2).

-implement also the discussion of results obtained with the identification of  fatty acid metabolism-related DEGs. Fatty acid metabolism is always cited but never discussed. Please Include a list of the main fatty acids related with the differentially expressed genes and comment on it. 

Replay: Thank you for your professional suggestion. We have added related content in discussion section according to your valuable suggestion(Line273-279, Line 299-305.We hope this content can meet with your comment.

Reviewer 2 Report

This is a descriptive study selecting 7 of 309 fatty acid metabolic genes from public databases to compute a risk score that is used to classify low- and high-risk groups for esophageal cancer and to correlate with clinical parameters. No validations in new patient population or biological analyses were conducted. Although the multiple databases were integrated in the computation, the risk score would be fluctuating depending on the arbitrary threshold and patient populations. The reproducibility of the seven fatty acid metabolic genes for predicting prognosis and immunotherapeutic response in esophageal cancer patients relies on the validations from a new patient population and/or biological assays.

Other issues:

1. The descriptions “role of fatty acid metabolism-related genes (FRGs) in ESCA has not been studied previously” (lines 12 to 13) and “present study is the first exploration of the relationship between genes involved in fatty acid metabolism and ESCA” (lines 264 to 265) are not accurate. Others have reported such genes, which can be found by Google Scholar search using the search terms “"esophageal cancer" AND "fatty acid metabolism" AND RNA-seq”.

2. The section 3.1, Fig. 1, and Table 1 were complied from public TCGA database and thus they should be presented in the Materials and Methods, not the Results section.

3. The number of patients in each comparison should be presented or annotated in the Figures for evaluation.

4. Please check typographical and grammatical errors carefully.

Author Response

This is a descriptive study selecting 7 of 309 fatty acid metabolic genes from public databases to compute a risk score that is used to classify low- and high-risk groups for esophageal cancer and to correlate with clinical parameters. No validations in new patient population or biological analyses were conducted. Although the multiple databases were integrated in the computation, the risk score would be fluctuating depending on the arbitrary threshold and patient populations. The reproducibility of the seven fatty acid metabolic genes for predicting prognosis and immunotherapeutic response in esophageal cancer patients relies on the validations from a new patient population and/or biological assays.

Replay: You have raised an important question. We are very grateful for your valuable comments. We understand that verification experiments may better improve the accuracy of our results. However, there are certain objective reasons that have influenced the progress of the experiment. For example, At present, there are relatively few patients with cervical cancer receiving immunotherapy, and there are certain limitations in the collection of clinical data and tissue specimens. Moreover, the design of primers, the purchase of related reagents and antibodies, and the collection of tissue specimens all require a certain amount of time, resulting in insufficient time. In addition, we have ordered related antibodies and are collecting related tumor and normal tissue specimens. Next, in the conclusion section (lines 363-365), we describe your question as a limitation of this research. We will perform further experiments to support our results. Thanks again for your constructive comments. We hope this content can meet with your comment.

Other issues:

  1. The descriptions “role of fatty acid metabolism-related genes (FRGs) in ESCA has not been studied previously” (lines 12 to 13) and “present study is the first exploration of the relationship between genes involved in fatty acid metabolism and ESCA” (lines 264 to 265) are not accurate. Others have reported such genes, which can be found by Google Scholar search using the search terms “"esophageal cancer" AND "fatty acid metabolism" AND RNA-seq”.

Replay: You have raised an important question. We are very grateful for your valuable comments. We have revised the manuscript carefully and add related content according to your suggestion (In Abstract section and in disscussion section, line 13-14, line 279-280 ).

  1. The section 3.1, Fig. 1, and Table 1 were complied from public TCGA database and thus they should be presented in the Materials and Methods, not the Results section.

Replay: This a very good point. The section 3.1, Fig. 1, and Table 1 is the analysis results based on the TCGA database, so the authors put this part at the results section. For example, Fig. 1A display 309 fatty acid metabolism-related genes from Kegg, Hallmark and Reactome by veen analysis. Fig. 1B and Fig. 1C display the differentially expressed FRGs based on Limma analysis. Table 1 display deatials of DE-FRGs. If reviewer feel it is more appropriate to put it in the Materials and Methods section, we will present the section 3.1, Fig. 1, and Table 1 in the Materials and Methods section according to your suggestion. We hope this content can meet with your comment.

  1. The number of patients in each comparison should be presented or annotated in the Figures for evaluation.

Replay: Thank you for your professional suggestion. We have ananoted the number of patients in the Materials and Methods section (line 71-72, line87-89, line121 ) and figure legend (line 170-171, line189 and line 249-250) according to your suggestion.

  1. Please check typographical and grammatical errors carefully.

Replay: Thank you for your comment. Our manuscript has undergone English language editing by MDPI. We will provide English-Editing-Certificate in our supplementary material. Thank again for your valuable comment.

Round 2

Reviewer 2 Report

1. Authors should state the impact of selection criteria other than the log2 fold change (FC) >1.5 in section 2.1. Identification of fatty-acid-metabolism-related genes.

2. The discrepancy in total number of esophageal cancer patients needs to be corrected. Line 70 indicates 160 tumors, but lines 85, 118, 166, and 244 state a total of 162 cancer patients. Figs. 7A-C show 184 primary tumors. Figs. 7D-G illustrate only 89 patients (23 high expression and 66 low expression).

3. Change the title “Table 2. Differentially expressed genes in high and low-risk groups” to “Table 2. Differentially expressed genes between high and low-risk groups”.

Author Response

Thank you for your professional suggestion.

  1. Authors should state the impact of selection criteria other than the log2 fold change (FC) ï¼ž5 in section 2.1. Identification of fatty-acid-metabolism-related genes.

Reply: You have raised an important question. We have added related content in Materials and Methods section (line 77-82) and delete the wrong discribe due to our carelessness. The selection criteria can be found in the relevant R language analysis code, if you have a need, we can provide the relevant original R code. We hope our reply can meet with your comments. If you have any question, please contact me. I appreciate you for your professional and helpful suggestion.

  1. The discrepancy in total number of esophageal cancer patients needs to be corrected. Line 70 indicates 160 tumors, but lines 85, 118, 166, and 244 state a total of 162 cancer patients. Figs. 7A-C show 184 primary tumors. Figs. 7E-G illustrate only 89 patients (23 high expression and 66 low expression).

Reply: Thank you very much for reviewing our manuscript carefully. I am so sorry for our mistake, the actual number of esophageal cancer patients was 160. We have corrected this mistake according to your suggestion. Moreover, Figs. 7A-C confirmation of the expression of FRGs in ESCA from UALCAN not only from TCGA, so Figs. 7A-C show 184 primary tumors. Figs. 7E-G Validation prognostic value of the seven DE-FRGs based on UALCAN database. Patients are mainly involved in esophageal adenocarcinoma (EAD) patients, not all esophageal cancer patients. So, Figs. 7E-G illustrate only 89 patients. Thank you again for your meticulous and professional review. We hope tha our reply can meet with your comment.

  1. Change the title “Table 2. Differentially expressed genes in high and low-risk groups” to “Table 2. Differentially expressed genes between high and low-risk groups”.

Reply: Thank your for your professional comment. We have change the title of Table 2 according your suggestion. The detail can be found in line 248 and Table 2. We hope this change can meet with your comment.
